# Postcode Lottery in Healthcare? Findings from the Scottish National Comprehensive Geriatric Assessment in Secondary Care Audit 2019

**DOI:** 10.3390/healthcare10010161

**Published:** 2022-01-14

**Authors:** Catriona Young, Alison I. C. Donaldson, Christine H. McAlpine, Marc Locherty, Adrian D. Wood, Phyo Kyaw Myint

**Affiliations:** 1School of Medicine, University of Aberdeen, Aberdeen AB25 2ZD, UK; 2School of Medicine and Dentistry, University of Aberdeen, Aberdeen AB25 2ZD, UK; alison.donaldson@abdn.ac.uk; 3NHS Greater Glasgow and Clyde, Glasgow G4 0SF, UK; Christine.McAlpine@ggc.scot.nhs.uk; 4School of Medicine, Dentistry & Nursing, University of Glasgow, Glasgow G12 8QQ, UK; 5NHS Grampian, Aberdeen AB25 2ZN, UK; marc.locherty@nhs.scot (M.L.); adrian.wood2@nhs.scot (A.D.W.); 6Ageing Clinical and Experimental Research Team (ACER), School of Medicine, Medical Sciences & Nutrition, University of Aberdeen & NHS Grampian, Aberdeen AB24 3FX, UK; phyo.myint@abdn.ac.uk

**Keywords:** care of the elderly, comprehensive geriatric assessment, health services, national audit

## Abstract

Comprehensive Geriatric Assessment (CGA) is provided differently across Scotland. The Scottish Care of Older People (SCoOP) CGA Audit was a national audit conducted in 2019 to assess this variation in acute hospitals. Two versions of audit questionnaires about the provision of CGA were developed (one each for larger hospitals and remote/rural areas) and piloted. The questionnaires were sent to representatives from all hospitals in Scotland using the REDCap (Research Electronic Data Capture) system. The survey asked each service to provide information on CGA service delivery at the ‘front door’. The questionnaire was open for completion between February and July 2019. Of the 28 Scottish hospitals which receive acute admissions, we received information from 26 (92.9% response rate). Reporting sites included seven hospitals from remote and rural locations in the Scottish Highlands and Islands. Significant variations were observed across participating sites for all key aspects studied: dedicated frailty units, routes of admission, staffing, liaison with other services and rehabilitation provision. The 2019 SCoOP CGA audit highlights areas of CGA services that could be improved and variation in specialist CGA service access, structure and staffing at the front door across Scotland. Whether this variation has an impact on the outcomes of older people requires further evaluation.

## 1. Introduction

The Scottish Care of Older People (SCoOP) national audit project was established with the overarching aim to improve care for older people in Scotland [1]. The vision is to provide a knowledge platform that can be built upon for a better understanding of standards of care, areas for improvement, and insight into the determinants for the best outcomes in care for older people, typically, yet not restricted to, those aged over 65. An initial scoping audit at Scottish Health Board level was completed in 2018, which showed that differences existed in acute and community service provision for older adults, and also in the consultant geriatrician provision per head of population aged over 65 [2].

The initial work of SCoOP demonstrated that we could work collaboratively in Scotland to evaluate and regularly monitor older people’s care through the SCoOP network and confirmed a need to look in more detail at how care is accessed by and provided for frail older adults in order to achieve excellence and equity in services across the nation. Frailty refers to an increased vulnerability to physiological stressors during a time of cumulative decline [3]. The focus of this report is on the delivery of comprehensive geriatric assessment (CGA) in acute hospitals across Scotland. CGA is a multidisciplinary, multi-dimensional diagnostic and therapeutic process that aims to determine the medical, mental, and functional problems of older people with frailty, thus facilitating the design of a coordinated and integrated plan for treatment and follow-up [4]. To date, research indicates that CGA should be the gold standard in providing care for older patients, with a Cochrane review in 2017 concluding that older patients are more likely to be alive and in their own homes at follow up if they receive CGA on hospital admission [5,6]. In a statement of intent [7], the Scottish government has committed to a multidisciplinary response for all older people requiring care.

Against this background, the objective of the nationwide survey was to identify variations in the structure and staffing of CGA services in acute hospital settings during 2019.

## 2. Materials and Methods

Scottish Health Board representatives on the SCoOP Steering committee (Appendix A) were asked to nominate representative/s from each hospital in their health board providing acute care to older adults who could answer questions about their local CGA provision. An audit questionnaire on CGA (Appendix A) was designed and piloted at 2 sites to gather information on how patients accessed consultant geriatrician-led care, the set-up for the care of frail older people at each hospital, the number of specialist staff employed, their working patterns, the frequency of multidisciplinary team (MDT) meetings and brief details of services linked to geriatric medicine that might influence the workload of the specialist geriatric medicine team, such as Hospital at Home, Orthogeriatrics and Surgical liaison services. The representative liaised with relevant staff, including service managers, the head of therapy services and others, to obtain accurate information for the hospital. A shortened set of questions was provided as an alternative for ‘remote and rural’ units where there was no or non-permanent geriatrician cover (Appendix A).

The project was commissioned by the Steering Group of the SCoOP (Appendix A) and approved by the British Geriatrics Society as part of the SCoOP National Audit Project.

The audit questionnaire was distributed using the REDCap system of the University of Aberdeen. REDCap is a secure web application supported by academic institutions around the world and compliant with the National Health Service (NHS) network requirements [8]. The questionnaire remained open for completion between February and July 2019; data were subsequently transferred into Excel, ‘sense checked’, and a descriptive analysis was performed. The data were presented descriptively: mean (standard deviation) and median (interquartile range) for normally and non-normally distributed continuous variables and number (percentage) for categorical variables.

## 3. Results

We identified 28 Scottish hospitals which receive acute admissions: of these, two did not respond and seven were located in remote and rural locations in the Scottish Highlands and Islands. The hospitals were anonymised by being assigned a number from ‘1 to 26’ and the health boards by being assigned a letter from ‘A to M’. The key findings for each aspect are summarised below. The detailed results are presented as a series of appendices.

### 3.1. Hospitals with and without Frailty Units

Seven hospitals (27%) reported having an acute frailty unit/service. Of these, four (57%) run a 24 h service 7 days a week (24/7). One (14%) operated only within normal working hours from Monday to Friday, and two hospitals (29%) have dedicated frailty assessment beds within a general medical ward area (Figure 1). The majority (86%) of these units used a form of frailty criteria as a screening tool. Age was another criterion used in the form of either greater than or equal to age 65 years (29%) or greater than or equal to 75 years (29%). A total of 43% used a combination of frailty and age to screen patients.

In the remaining 19 hospitals, patients admitted with frailty received their initial assessment in either a general medical admissions unit (36%), general medical ward (42%) or a general ward for older adults (21%).

### 3.2. Routes of Admission

All hospitals with consultant geriatrician cover were asked about all possible routes of admission under their care. The most frequent route was via an acute medicine department (48%) or by patients identified by a geriatrician in the acute medicine department (38%); other routes included the emergency department (35%) and direct general practitioner (GP) referrals (17%). Further details for those units that do not fit the criteria of these categories are included in Appendix A. Of note, several hospitals had more than one route into consultant geriatrician-led care.

### 3.3. Specialist Staffing

#### 3.3.1. Senior Medical Staffing

There is a wide variation in the size of the population aged ≥65 years served by each health board, ranging from ~5000 to ~200,000 (Table 1). Similarly, the number of full-time equivalent (FTE) geriatric medicine consultants per 10,000 older people aged ≥65 years ranges across the Scottish health boards, with a median of 1.27 (range: 0.0–2.27) (Table 1).

There is also variation in the number of acute sessions worked by geriatricians in each health board, with a median of 1.1 (range: 0.0–3.7). Geriatricians based in hospitals 4 and 21 had no sessions specifically for the admissions of frail older adults at the time of the audit, whereas the highest number was reported by hospital 7 (mean 3.7 acute sessions). The mean number of hours per weekday (Monday–Friday) spent by consultants reviewing new patients was 4.6 h and 3.1 h at the weekend (Saturday and Sunday). Without including remote and rural hospitals, the mean hours per weekday for consultants reviewing new patients were 5 h and 3.6 h at the weekend.

#### 3.3.2. Physiotherapists

The questionnaire asked about the number of therapists specifically employed to review patients admitted acutely with frailty. The number of specialist physiotherapists per 10,000 older people aged ≥65 years varies across the Scottish Health Boards with a median of 0.22 (range: 0.0–1.0) physiotherapists dedicated to acute geriatric medicine per 10,000 population aged ≥65 years. The mean hours spent by physiotherapists reviewing new admissions to geriatric medicine beds were 4.6 h during weekdays (Monday to Friday) and 1.8 h at the weekend.

#### 3.3.3. Occupational Therapists

The median number of specialist occupational therapists per 10,000 population aged ≥65 years was 0.34 (range: 0.0–2.13). Similar to that of physiotherapists, the mean hours spent by occupational therapists reviewing new admissions to geriatric medicine beds were 4.7 h during weekdays and 1.7 h at the weekend.

We were unable to correlate the number of frailty specific therapists in each unit with the size of the older patient population covered, but there are eight hospitals without frailty specific physiotherapists and occupational therapists, of which only four centres are remote and rural.

### 3.4. Access to Liaison Psychiatry

Psychiatrists were based in the admission setting or routinely available to review newly admitted frail patients in five hospitals (19%), and most units were able to request a review by a psychiatrist for their frail patients, but this did not usually occur on the same day as requested (69% could access psychiatry review on request, but this was not usually conducted on the same day) (Table 2).

### 3.5. Pharmacists

At eighteen hospitals, acutely admitted patients to geriatric wards (69%) will have their medications reviewed by a pharmacist within 24 h, either by a pharmacist employed specifically for CGA (*n* = 6) or shared with other units (*n* = 12).

### 3.6. Social Work

Seven hospitals answered yes (27%) to the question of if a member of a social work team would usually be available to review a patient within 24 h of admission.

### 3.7. Bed Management

One hospital (4%) had a dedicated bed manager for the geriatric medicine service.

### 3.8. Multidisciplinary Team Meetings

MDT meetings were held in 65% of hospitals at least once daily during the week. However, MDT meetings were also held at the weekend in only four units (15%). The most consistent members of the MDT involved in meetings were consultants, physiotherapists and occupational therapists. Multidisciplinary teams shared notes in 15 hospitals (58%).

### 3.9. Additional Services

#### 3.9.1. Hospital at Home

Of all the hospitals, 42% provided a form of the ‘hospital at home’ service, seen in six of the Scottish Health Boards (see Table 3).

#### 3.9.2. Orthogeriatrics

There were 12 hospitals who identified themselves as providing active input into orthopaedics (see Table 4).

#### 3.9.3. Surgical Liaison

There were five Scottish hospitals (19%) with scheduled input into older patients under care by surgical specialities, a further two centres described an arrangement of case-by-case referral (Table 5).

## 4. Discussion

To the best of our knowledge, this is the first national report of the provision of CGA care for older adults acutely admitted to hospital. We observed considerable variations in how CGA services are set up and staffed, and thus in how comprehensive geriatric assessments are conducted. While understanding the true impact of this on patient outcomes will only be appreciable once these data are available for correlation with individual patient-level outcome data, it is already apparent that there may be inequity in service provision for frail older adults. This audit highlights where individual departments may have useful expertise and experience in service development, such as in running an acute frailty unit or delivering Hospital at Home. Further collaborative working by sharing this expertise and experience and learning from regular audits should help us to improve acute care for frail older adults in Scotland and shape Scottish geriatric medicine into a world-leading service.

One of the main challenges of this audit was overcoming the many differences in how services are set up, to enable comparison. At the time of this audit in 2019, only 27% of Scottish hospitals had a frailty unit, whereas the NHS England Benchmarking Network reported that 52% of trusts had a frailty unit in 2016 [9]. Even among hospitals with acute frailty units, there are differences in the service they provide, with some offering an alternative ‘front-door’ to acute medicine or the emergency department and others limited to providing assessment only within normal working hours. Currently, the quality of evidence for admission through frailty assessment units is of low quality, albeit with suggested benefits in reducing readmissions and costs [10].

Overall, in Scotland, admissions into specialist geriatric care are mostly routed via an acute medicine or emergency medicine department, with only 17% of units accepting direct admissions from primary care. There are clear benefits from frail patients being admitted directly into specialist care. However, one in three of the Scottish population aged over 75 was admitted at least once to hospital in 2017/18, and there will be a predicted 19% increase in the population aged over 65 over the next decade [11,12]. Geriatric medicine specialists will need to focus their expertise on those who are frail, rather than all those in a certain age bracket, with significant implications on the demand for geriatric medical care. Indeed, this audit shows that Scottish geriatric medicine services are already aligned more to criteria of frailty than age for accepting patients.

We recognise the limitations of drawing comparisons of the specialist geriatric medicine workforce based on the Information Services Division data on the population of over 65 yearolds as a denominator rather than the number of frail individuals in a specific catchment area, but such information is not currently available. Clearly, the health of the population, rather than age, will shape the demand for these services, and we know that the life expectancy among Scottish NHS Health Boards varies considerably (79.7 years for a male in one of the islands compared to 74.5 years in the largest city in 2010–12), but also within health boards (there is a range of 72.6 years to 80.1 years in the most populated health board) [13]. Despite this, with a median of 1.27 full-time equivalent consultant geriatricians per 10,000 people aged over 65 years in Scotland with a wide range from 0 to 2.27, there is clearly a need to look into this postcode lottery in access to specialist care, especially in relation to rurality. The survey highlights an inequality of services in terms of rurality without insight into how deprivation might also be contributing to a postcode lottery.

Some of the inequities in the division of resources amongst Scottish Health Boards appear to be influenced by the concentration of the population geographically within Scotland’s ‘Central Belt’. Difficulties with recruitment to the more ‘remote and rural’ areas may contribute, but even if all vacancies at the consultant level at the time of the audit were filled, the disparities would not be fully addressed. Another interesting finding is the variation in the number of ‘acute’ sessions worked by consultant geriatricians, suggesting the job plans between different health boards are quite variable with some hospitals perhaps able to invest more consultant time into community care and admission avoidance than others. This alone would be an interesting topic for a future audit, especially with regard to the influence on consultant recruitment and job satisfaction.

Another challenge for this audit was assessing the availability of staff and services which may be shared between departments, especially in smaller hospitals. The results do indicate a variation in specialist therapist provision between hospitals, but also show a reduction in therapist availability at the weekend. This trend was also seen with the number of hospitals holding daily MDTs at the weekend, just 15% compared to 65% during the normal working week. It will be interesting to explore if the day of admission has any bearing on outcome measures once Information Services Division data is available.

This audit asked for a brief description of additional services provided by departments, and these will be helpful in guiding future audit cycles. Certainly, it will be interesting to note the progression of the Hospital at Home services since this audit was conducted in 2019. The experience and adaptation of geriatric services over the last year with the influence of COVID-19 on service pressures will undoubtedly have led to changes in the results we present here, and we must recognise this when interpreting them.

Correlation with outcome data is needed to ascertain the influence of these identified differences on patient care. Therefore, further data collection and research here is required.

Despite these limitations, this is the first report of its kind with regard to CGA provision to hospitalised older frail patients.

## 5. Conclusions

CGA in Scottish Hospitals highlights variation in the ways acute comprehensive specialist care is accessed, structured and staffed across the country. Our findings provide essential information for clinicians, service providers, policy makers and the public to improve their local services. Therefore, our results should offer a basis for opening discussion between services to learn from each other’s expertise as we aim to work collaboratively to improve acute care for frail older adults in Scotland and shape Scottish Geriatric Medicine into a world-leading service.

## Figures and Tables

**Figure 1 healthcare-10-00161-f001:**
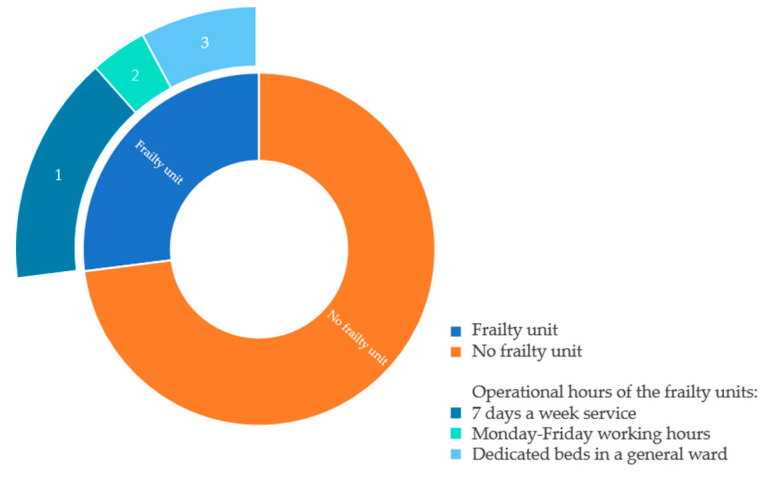
Operational hours of the frailty units.

**Table 1 healthcare-10-00161-t001:** NHS Scottish Health Board population and the full-time equivalent geriatrician provision per 10,000 people ≥65 years (based on 2019 mid-year population estimates).

NHS Health Board	Population Number ≥ 65 Years	FTE Geriatrician per 10,000 People ≥65
A	4686	0
B	6895	0
C	84,228	0.62
D	38,570	0.65
E	53,088	0.89
F	107,946	1.13
G	77,024	1.27
H	89,691	1.56
I	28,616	1.75
J	59,174	2.2
K	121,805	2.22
L	195,952	2.27
M	148,954	2.29

Note: Data presented are at health board level.

**Table 2 healthcare-10-00161-t002:** Availability of Liaison Psychiatry.

Availability of Liaison Psychiatry	Number of Hospitals
No access on site	3
Review within hospital on request, not usually same day	18
Review within hospital on request, usually same day	4
Psychiatrist based in admission setting	1

**Table 3 healthcare-10-00161-t003:** Hospital at Home services as described by geriatricians based at each centre.

Health Board	Hospital Code	Description of Service
**C**	1	No medical staff attached but is described as an ‘enhanced intermediate care team’ with ANP ^a^ and AHPs ^b^.
	2	Community rehab team and alternatives to admission, by supporting GP; no geriatrician involved; but ANP is.
**G**	5	Three bases throughout the health board. Admits step-down patients from acute hospital, to facilitate discharge, and takes direct referrals from GPs. 40–60 patients on virtual ward round on a given day across 3 bases. Consultant virtual ward round 2 or 3 times per week, with middle grade (FY ^c^ 2 or above) on other weekdays, and nurses at weekends.
**F**	8	Not truly Hospital at Home but urgent domiciliary visits can be arranged to try and prevent admissions. These are complete by consultant and ANPs.
	7	Current trial of “acute care at home” service—ANP, AHPs, HCSWs ^d^—offering admission avoidance support with medical input from GPs in part of the city. No regular geriatrician input to this service.
**K**	15	Hospital at Home team providing multidisciplinary care at home.
	13	Consultant/nursing/OT/PT takes hospital referrals 7 days/week from community on normal working days. SAS ^e^ referrals at weekend.
**M**	18	Three Hospital at Home service (in 3 out of the 5 areas within the health board).
	16	Hospital at Home team sees around 100 patients per month with length of stay around 4 days.
	17	Hospital at Home services.
**H**	20	Hospital at Home and ECS ^f^ (enhanced community support) for patients earlier in the journey to prevent crisis. More wrapped around GP practices.

^a^ Advanced Nurse Practitioners, ^b^ Allied Health Professionals, ^c^ Foundation Year, ^d^ Healthcare Support Worker, ^e^ Specialist Associated Service and ^f^ Enhanced Community Support.

**Table 4 healthcare-10-00161-t004:** Orthogeriatrics service as described by geriatricians based at each centre.

Health Board	Hospital Code	Description of Service
**C**	2	Two sessions per week by consultant geriatrician.
**G**	5	There were 3.5 consultant sessions per week. All hip fracture patients reviewed, and other patients by request from orthopaedic team. Staff Grade Orthogeriatrician works 6 sessions per week.
**F**	8	Consultant ward round Mon and Wed morning. ANP ^a^ input Mon-Fri morning who review all new hip fractures. No service at weekend.
	7	Twice weekly ward rounds. Support of “orthogeriatric” ANPs.
**L**	10	Consultant cover 5PAs; ECON ^b^ 3.8wte working across hospital with 7 days cover.
	9	Two DCC ^c^ of consultant time with 2 consultants doing a ward round in trauma wards i.e., X2 visits in total per week. Orthopaedic ECONS collect fairly detailed information on the patients before ward rounds and provide daily input to the wards.
**K**	15	Daily input with Consultants and ACE ^d^ nurse Mon–Friday and ACE nurse alone Sat–Sunday. Consultants will review patients at request of ACE nurses at weekends.
	13	Daily input Monday–Friday via ACE nurse and Consultant Geriatrician.
**M**	18	Orthogeriatric rehabilitation service only—trauma managed initially at another hospital in the same city.
	16	Combined orthopaedic and general rehab ward (30 beds).
	17	Two trauma wards. Two consultant ward rounds and one MDT per week from two consultants aligned to each trauma ward. Daily SD ^e^ and Clinical fellow input to support orthopaedics. Weekdays only. Consultant Geriatrician on call at weekend can provide input if requested.
**H**	20	Run by SD, team of clinical fellows and FY2 from orthopaedics.

^a^ Advanced Nurse Practitioner, ^b^ Elderly Care Assessment Nurses, ^c^ Direct Clinical Care, ^d^ Acute Care of the Elderly nurses and ^e^ Specialty Doctors.

**Table 5 healthcare-10-00161-t005:** Surgical liaison services for frail patients.

Health Board	Hospital Code	Description of Service
**L**	10	No nurse support in 2019, specialty doctor, 2 Pas ^a^, Consultant 0.5 Pas
	9	0.5 DCC ^b^ consultant session weekly—they conduct a weekly MDT and see appropriate patients after that.
**K**	15	Case-by-case referral.
	13	Once weekly ward round—patients selected by surgical team for review.
**M**	18	Surgical liaison service and POPs ^c^ service led by a geriatrician with nurse support. A POPs clinic also operated with surgeons and anaesthetic colleagues to determine better outcomes for patients with frailty pre- and post-surgery.
	16	Case-by-case referral.
	17	Two consultant ward rounds and one MDT with three consultants delivering POPS service. Daily weekday input from one speciality doctor. Specialist nurse.

^a^ Physician associates, ^b^ Direct Clinical Care and ^c^ Proactive care of older people undergoing surgery.

## Data Availability

The data presented in this study are available in Appendix A.

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
