# Peer review of "Postcode Lottery in Healthcare? Findings from the Scottish National Comprehensive Geriatric Assessment in Secondary Care Audit 2019"

_healthcare, 2022, doi:10.3390/healthcare10010161_

Round 1

Reviewer 1 Report

Postcode lottery in healthcare? Findings from the Scottish National Comprehensive Geriatric Assessment in Secondary Care Audit 2019

The topic is interesting and address how Comprehensive Geriatric Assessment (CGA) is provided differently across Scotland depending on whether a person lives rurally or remotely. While the manuscript is thought-provoking, there is a need for a major revision to warrant publishing.

Abstract: line 19: please define RED then use the abbreviation.

Introduction: lines 32-54 is lacking a sufficient background of the needs and definitions introduced later in the manuscript. For example you do not define your population; clarify older people. What is this in Scotland, age cut off? Additionally, you introduce frailty in line 79, but have not defined it in your background. Line 34: remove website and add reference correctly in endnote/references. Line 53-54: recommend revising your aim of the study. Rewrite for the scientific population. An aim usually starts with “to determine…” for example …if CGA services are dependent upon post code…or if CGA services are dependent upon rural versus remote location.  (Prior to your revising your aim it might also be helpful for the reader to know in the background if there are any laws or acts in the government related to health care services for older people, that they should be equal, for example.  Alternatively, you can write an objective, but you need to be clear for the reader.  

Method: disseminated is not a method rather this is an aim or goal, for example to spread or give out something, especially news, information or ideas to many people. Recommend rewriting this to be clear for the reader that descriptive analysis was reported. Did you consider looking at odd ratio (OR) comparing rural and remote results? Could be of interest. Or p value? To compare the two groups, since you mention this as a result in your title.  

Results: The manuscript I received to review is incomplete. There are no appendices to reference, which is a weakness. I contacted the editor and they said they were not included when the manuscript was submitted. Based on this I find the results and data to be somewhat confusing as it is currently presented to the reader.  You are not consistent in reporting your data either (examples from line 80, 85 and 134), sometimes you report the number, other times the fraction and sometimes both or just the percent. Please be consistent.  Since I am unable to see you appendices, if you haven’t already done so consider presenting this data is tables, so make it easier for the reader to understand your results.

Line 80: 7/26? Aren’t there 28 hospitals?  Either check your data or if you have missing data you have to tell the reader this in a footnote.

Line 75: be objective, no need to add the word only in your result reporting. Save that for the discussion.

Line 91 recommend changing language to be more scientific or use medical terms. “into a bed”, do you mean admission into hospital?   In table 1, you report data for 65+, but you mentioned earlier 75+, are you combining all the data here, or are the 75+ groups not considered, since you are only discussing data only from 13 hospitals. You might consider clarifying or specifying if data is missing.

There are multiple cases of the authors using abbreviations without clarifying earlier in the text what they are. RED Cap, FTE, GP, POP’s, ACE, GGH, DCC… it could be that the appendices are missing or that it is an oversight on the authors. Please check the entire manuscript for these issues. The readers should not need to look for the meanings outside of the manuscript, and please ensure the table are self-explanatory, as they should be able to stand for themselves, without the manuscript.  

Line 141: Seven hospitals (27%) answered yes (keep your data close to the topic).

Line 145: One hospital (3.6%) had a dedicated… (remove the word only, this is subjective)

Line 147-148: Multidisciplinary team meetings were held in 65% of hospitals… (remove most), add MDT directly after you introduce it in the text.

Section 3.7 consider combining with 3.22 (therapy staff), since you are discussing the same healthcare professionals.

3.8.1 39% of hospitals provide at home services…The services vary in form... see table 3.

3.8.2 What about therapy staff statistics related to his topic? Should address. If no information was collected, consider adding this as a weakness in the discussion where data collection did not consider the entire team…  

Line 187: this is not how you reference, use endnote and read guidelines from Vancouver how to handle this type of reference.

Line 191-197: consider revising language and divide into two sentences for the reader too long otherwise.

Line 200: clarify ISD data  

Line 204-205 You confuse the reader by adding this detailed information in your discussion. Recommend you make it more general and rather than discuss specific townships, cities, discuss in terms of rural versus remote. Remove the names of the towns.

Line 209: related to access of services determinat upon rural and remote locations throughout Scotland.

Conclusion:

Line 237: suggest removing line 237. Start conclusion with:  CGA in Scottish hospitals varies greatly….

Line 239-240: remove “correlation with outcome data is a weakness or limitation”. It does not belong in your conclusion. Move to discussion, weakness or remove entirely.

Consider adding further weaknesses to your study: surveys were done by one person in  the hospital and results may be subjective as to how that administrator experienced their organization in providing care to the older population.

Author Response

21/12/2021

Subject: Re-submission of Manuscript ID: healthcare-1520010

Dear Sir/Madam,

I would like to express our appreciation for the feedback that you provided for our paper. The reviewer’s feedback has significantly improved our paper.

Thank you for the opportunity to resubmit “Postcode lottery in healthcare? Findings from the Scottish National Comprehensive Geriatric Assessment in Secondary Care Audit 2019” for further consideration in Healthcare.                                                                                                                                              

Please find below our responses to the reviewers’ comments. We have thoroughly revised the manuscript, addressing the comments with the recommended appropriate changes. We have amended the manuscript seen below and in attached documents to address these concerns (additions in blue, deletions in red, please see attachment).

We sincerely hope that the revised manuscript addresses all the issues.

We look forward from hearing from you.

Yours sincerely,

Catriona Young (on behalf of the authors)

Under abstract:

Comments to the authors:  line 19: please define RED then use the abbreviation.

Response: Thank you for raising this issue of abbreviation use as well as the others found in the paper. We have expanded on this abbreviation, REDCap (Research Electronic Data Capture) (line 19-20).

Under Introduction:

Comments to the authors:  lines 32-54 is lacking a sufficient background of the needs and definitions introduced later in the manuscript. For example you do not define your population; clarify older people. What is this in Scotland, age cut off? Additionally, you introduce frailty in line 79, but have not defined it in your background.

Response: We have amended at the end of the sentence “The vision is to provide a knowledge platform that can be built upon for better under-standing of standards of care, areas for improvement, and insight into what the determinants are for best outcomes in care for older people, typically yet not restricted to those aged over 65.” (Line 38).

We have also addedFrailty refers to an increased vulnerability to physiological stressors during a time of cumulative decline [3].” (Lines 45-47) ([3] Clegg A., Young J., Illiffe S., Rikkert MO. & Rockwood K. 2013, “Frailty in elderly people”. The Lancet 2013; vol. 381; pp. 752-762.)

Comments to the authors:   remove website and add reference correctly in endnote/references.

Response: Thank you for this observation. We have corrected it with Vancouver referencing.

Comments to the authors:  Line 53-54: recommend revising your aim of the study. Rewrite for the scientific population. An aim usually starts with “to determine…” for example …if CGA services are dependent upon post code…or if CGA services are dependent upon rural versus remote location.  (Prior to your revising your aim it might also be helpful for the reader to know in the background if there are any laws or acts in the government related to health care services for older people, that they should be equal, for example.  Alternatively, you can write an objective, but you need to be clear for the reader.

Response: We have changed those lines from “Against this background we carried out a nationwide survey focusing on “who, what, when and how” CGA services were delivered in acute hospital settings during 2019”  to Against this background, the objective of the nationwide survey was to identify variations in the structure and staffing of CGA services in acute hospital settings during 2019.” (Line 57-58).

We appreciate how the comment to relate CGA in Scotland to law/acts would be useful and relevant. We had difficulty connecting it to a law, in its replacement we have now mentioned the Scottish government’s Health and social care for older people: statement of intent (March 2021). We added, “In a statement of intent [7], the Scottish government has committed to a multidisciplinary response for all older people requiring care.” (Lines 54-56)

Under methods:

Comments to the authors:  disseminated is not a method rather this is an aim or goal, for example to spread or give out something, especially news, information or ideas to many people. Recommend rewriting this to be clear for the reader that descriptive analysis was reported.

Response: Thank you for highlighting this. We have replaced “disseminated” with “distributed” (line 73).

Under results:

Comments to the authors:   Did you consider looking at odd ratio (OR) comparing rural and remote results? Could be of interest. Or p value? To compare the two groups, since you mention this as a result in your title.

Response: This is really a valid point and we thank the reviewer for raising this. As can be seen, there is no systematic and uniform delivery of services across the Scotland with regards to CGA provision by Older People’s Services. We acknowledge the variation may exist for a very good reason given the local issues such as availability of other services. Therefore, it is not possible to statistically compare results. We have therefore taken the descriptive approach which will have impact on local services to understand where improvements are required in the context of their local population needs and staffing level/resources. We believe our work will be a key driver in improving services by sharing the information across Scotland, as well as exemplar for other global service evaluation work in healthcare of older people.

Comments to the authors: The manuscript I received to review is incomplete. There are no appendices to reference, which is a weakness. I contacted the editor and they said they were not included when the manuscript was submitted. Based on this I find the results and data to be somewhat confusing as it is currently presented to the reader.

Response: We profusely apologize about this oversight. All 12 appendices should have been uploaded. We have uploaded all appendices in this revised submission.

Comments to the authors:  You are not consistent in reporting your data either (examples from line 80, 85 and 134), sometimes you report the number, other times the fraction and sometimes both or just the percent. Please be consistent.  Since I am unable to see you appendices, if you haven’t already done so consider presenting this data is tables, so make it easier for the reader to understand your results.

Response: Thank you for pointing this out. The fractions were mentioned when the denominator was the number of hospitals with frailty units rather than all hospitals in case readers felt there was a miscalculation. However, we can see it makes the reporting appear inconsistent. All the values are now presented as percentages.

Comments to the authors: 7/26? Aren’t there 28 hospitals?  Either check your data or if you have missing data you have to tell the reader this in a footnote.

Response: We thank the reviewer for this comment which we agree needs further explanation. There are 28 hospitals across Scotland which has care of the elderly services however only 26 responded to our survey. We have added at the start of the results, “We identified 28 Scottish Hospitals which receive acute admissions: of these 2 did not respond and 7 were located in remote and rural locations in the Scottish Highlands and Islands.” (Lines 80-81).

Comments to the authors:  Line 75: be objective, no need to add the word only in your result reporting. Save that for the discussion.

Response: Thank you. We have removed the use of “only”.

Comments to the authors: Line 91 recommend changing language to be more scientific or use medical terms. “into a bed”, do you mean admission into hospital?

Response: We thank the reviewer for their comment. We have changed that first sentence in now section 3.2  from “All hospitals with consultant geriatrician cover (n=23) were asked about all possible routes of admission for a frail patient into a bed under the care of a geriatrician” to “All hospitals with consultant geriatrician cover (n=23) were asked about all possible routes of admission under their care.” (Line 98-99).

Comments to the authors: In table 1, you report data for 65+, but you mentioned earlier 75+, are you combining all the data here, or are the 75+ groups not considered, since you are only discussing data only from 13 hospitals. You might consider clarifying or specifying if data is missing.

Response: Thank you for pointing out this inconsistency. With the exception of individual hospitals using +75 year as their admission criteria seen in appendices 4 and 5. We have kept it to +65 years old for consistency.

The reason there is 23 hospitals (line 98) with consultant geriatrician cover and 13 rows in Table 1 is because each row represents a health board rather than hospital. We were unable to get the population that each hospital serves and hence presented the data at Health Board level.  We have added a footnote “Note: Data presented are at Health Board level” under the table to highlight this.

Comments to the authors: There are multiple cases of the authors using abbreviations without clarifying earlier in the text what they are. RED Cap, FTE, GP, POP’s, ACE, GGH, DCC… it could be that the appendices are missing or that it is an oversight on the authors. Please check the entire manuscript for these issues. The readers should not need to look for the meanings outside of the manuscript, and please ensure the table are self-explanatory, as they should be able to stand for themselves, without the manuscript.

Response: We have overlooked the abbreviations presented in the tables, thank you for identifying these among others in the text. We have added explanatory footnotes below tables and brackets following terms in the main text:

  • REDCap (Research Electronic Data Capture) (line 19-20)
  • MDT (Multi-disciplinary Team) (line 66)
  • NHS (National Health Service) (line 75)
  • GP (General Practitioner) (line 101)
  • FTE (Full Time Equivalent) (line 109)
  • Footnotes for table 3: aAdvanced Nurse Practitioners, bAllied Health Professionals, cFoundation Year, dHealthcare Support Worker, eSpecialist Associated Service and fEnhanced Community Support.
  • Footnotes for table 4: aAdvanced Nurse Practitioner, bElderly Care Assessment Nurses, cDirect Clinical Care, dAcute Care of the Elderly nurses and eSpecialty Doctors.
  • Footnotes for table 5: aPhysician associate, bDirect Clinical Care and cProactive care of older people undergoing surgery.
  • ISD (Information Services Division) (line 216)

Comments to the authors: Line 141: Seven hospitals (27%) answered yes (keep your data close to the topic).

Response: We have amended as “Seven hospital answered yes (27%)” (Line 86).

Comments to the authors: Line 145: One hospital (3.6%) had a dedicated… (remove the word only, this is subjective)

Response: We have removed the use of “only”.

Comments to the author: Line 147-148: Multidisciplinary team meetings were held in 65% of hospitals… (remove most), add MDT directly after you introduce it in the text.

Response: We have removed the use of “most” as suggested by the reviewer and introduced MDT as an abbreviation earlier in the methods. The sentence now reads as MDT meetings were held in 65% of hospitals at least once daily during the week.” (Line 156).

Comments to the authors: Section 3.7 consider combining with 3.22 (therapy staff), since you are discussing the same healthcare professionals.

Response: We thank the reviewer for this comment. We feel the MDT membership extends beyond medical/PT/OT/nursing staff and those listed between 3.1 and 3.7 should ideally be part of MDT membership in delivering CGA. Hence, we have placed MDT team at what is now 3.8.

Comments to the authors: 3.8.1 39% of hospitals provide at home services…The services vary in form... see table 3.

Response: Thank you for the feedback. We appreciate how it is better to introduce Hospital at Home as a percentage. We have presented as “42% of hospitals also provided Hospital at Home services”. (Lines 163-164).

Comments to the authors: 3.8.2 What about therapy staff statistics related to his topic? Should address. If no information was collected, consider adding this as a weakness in the discussion where data collection did not consider the entire team… 

Response: Our focus is on contribution of geriatric services for orthopaedics liaison (3.9.2, header numbering changed as the result of comments from the other reviewer) purpose and therefore we do not collect any information of therapy input managed by Orthopaedics department. Scottish Hip Fracture Audit provides such information and this is beyond the scope of our work. We have clearly presented team composition of ortho liaison service (where present) in Table 4.

Comments to the authors: Line 187: this is not how you reference, use endnote and read guidelines from Vancouver how to handle this type of reference.

Response: We apologize for this oversight. We have corrected referencing in line with author guidelines.

Comments to the authors: Line 191-197: consider revising language and divide into two sentences for the reader too long otherwise.

Response: We have broken the lengthy sentence in question into 3 sentences “There are clearly benefits from frail patients being admitted directly into specialist care. However, one in three of the Scottish population aged over 75 are admitted at least once to hospital in 2017/18 and there is a predicted 19% increase in the population aged over 65 over a decade [10, 11]. Geriatric medicine specialists will need to focus their expertise on those with decompensated frailty, rather than all those in a certain age bracket, with significant implications on demand for geriatric medical care.” (Lines 207-213).

Comments to the authors: Line 200: clarify ISD data

Response: We have explained this abbreviation as Information Services Division. (Line 216)

Comments to the authors: Line 204-205 You confuse the reader by adding this detailed information in your discussion. Recommend you make it more general and rather than discuss specific townships, cities, discuss in terms of rural versus remote. Remove the names of the towns.

Response: We have changed this from “Clearly the health of the population, rather than age, will shape the demand for ser-vices, and we know that the life expectancy among Scottish NHS Health Boards varies considerably (79.7 years for a male in Orkney compared to 74.5 years in Greater Glasgow in 2010-12), but also within Health Boards (80.1 years for a male in East Dunbartonshire compared to 72.6 years for Glasgow City - both areas within NHS Greater Glasgow and Clyde) [10].” to “Clearly the health of the population, rather than age, will shape the demand for services, and we know that the life expectancy among Scottish NHS Health Boards varies considerably (79.7 years for a male in one of the islands compared to 74.5 years in the largest city in 2010-12), but also within Health Boards (there is a range of 72.6 years to 80.1 years in the most populated Health Board) [10].”(lines 218-222) we have also removed Aberdeen” earlier in the text (line 116).

Comments to the authors: Line 209: related to access of services determinant upon rural and remote locations throughout Scotland.

Response: We added on lines 225-226 directly after the sentences in the previous response, “Despite this, with a median of 1.27 full time equivalent consultant geriatricians per 10,000 people aged over 65 years in Scotland with a wide range from 0 to 2.27, there is a clearly a need to look into this postcode lottery in access to specialist care, especially in relation to rurality.”

Under conclusion:

Comments to the authors: Conclusion: Line 237: suggest removing line 237. Start conclusion with:  CGA in Scottish hospitals varies greatly….

Response: We removed “This initial presentation of the results of the 2019 SCoOP audit of the provision of” and it now reads “CGA in Scottish Hospitals highlights variation in the ways acute comprehensive specialist care is accessed, structured and staffed across the country.” (Lines 257-258)

Comments to the author: Line 239-240: remove “correlation with outcome data is a weakness or limitation”. It does not belong in your conclusion. Move to discussion, weakness or remove entirely.

We removed “Correlation with outcome data is needed to ascertain the influence of these differences on patient care” from the conclusion. We now stated “Correlation with outcome data is needed to ascertain the influence of these identified differences on patient care.  Further data collection and research here is required” in lines 252-253 as a separate paragraph before “Despite these limitations …”.

Comments to the author: Consider adding further weaknesses to your study: surveys were done by one person in  the hospital and results may be subjective as to how that administrator experienced their organization in providing care to the older population.

Response: We apologise for slight inaccuracy in our presentation which led to this confusion. In fact, the representative acted as lead data collector and they liaised with relevant staff including service managers, head of therapy services etc. to obtain accurate information. This has now been added to the methods as, “The representative liaised with relevant staff including service managers, head of therapy services and others to obtain accurate information for the hospital.” (Lines 69-70).

Reviewer 2 Report

Well done and very interesting.

Minor changes only.

Explain the joke: what do you mean by postcode lottery

Line 79: this statement has no context. Is it a header or is tit misplaced?

lines 80-88: could you create an infographic that shows these result, perhaps in a pie chart?

Table 3: you have used several acronyms that are not defines. They include ANP, AHP, QEUH, POPS, MDT, HSCWs ISD, NHS, GGH GORU, etc. You need to help out those not lucky enough to be Scottish and define these acronyms.

Author Response

21/12/2021

Subject: Re-submission of Manuscript ID: healthcare-1520010

Dear Sir/Madam,

I would like to express our appreciation for the feedback that you provided for our paper. The reviewer’s feedback has significantly improved our paper.

Thank you for the opportunity to resubmit “Postcode lottery in healthcare? Findings from the Scottish National Comprehensive Geriatric Assessment in Secondary Care Audit 2019” for further consideration in Healthcare.             

Please find below our responses to the reviewers’ comments. We have thoroughly revised the manuscript, addressing the comments with the recommended appropriate changes. We have amended the manuscript seen below and in attached documents to address these concerns (additions in blue, deletions in red, please see the attachment).

We sincerely hope that the revised manuscript addresses all the issues.

We look forward from hearing from you.

Yours sincerely,

Catriona Young (on behalf of the authors)

Comments to the author: Explain the joke: what do you mean by postcode lottery

Response: We apologise for not providing a definition of  'postcode lottery' : this phrase means 'variations in health care between different geographical areas  that appear arbitrary and unrelated to health needs'.  We would like to specifically draw attention to the post code lottery issue. It was very helpful to realize we had not yet related the paper back to the idea of ‘postcode lottery’. We added “The survey highlights an inequality of services in terms of rurality without an insight about how deprivation might also be contributing to a postcode lottery” to line 226 as a part of the same paragraph.

Comments to the author: Line 79: this statement has no context. Is it a header or is it misplaced?

Response: We thank the reviewer for highlighting this oversight. We apologise for the error. This has now been corrected by making it a header and adjusted the numbering of the subsequent headers.

Comments to the author: lines 80-88: could you create an infographic that shows these result, perhaps in a pie chart?

Response: we appreciate this suggestion of creating a pie-chart about frailty units and their service. This has now been included as Figure 1 below (line 93).

Figure 1. Operational hours of the frailty units

Comments to the authors: you have used several acronyms that are not defines. They include ANP, AHP, QEUH, POPS, MDT, HSCWs ISD, NHS, GGH GORU, etc. You need to help out those not lucky enough to be Scottish and define these acronyms.

Response: Thank you for noticing the tables full of unexplained abbreviations that we overlooked. We have added explanatory footnotes below tables and brackets following terms in the main text:

  • REDCap (Research Electronic Data Capture) (line 19-20)
  • MDT (Multi-disciplinary Team) (line 66)
  • NHS (National Health Service) (line 75)
  • GP (General Practitioner) (line 101)
  • FTE (Full Time Equivalent) (line 109)
  • Footnotes for table 3: aAdvanced Nurse Practitioners, bAllied Health Professionals, cFoundation Year, dHealthcare Support Worker, eSpecialist Associated Service and fEnhanced Community Support.
  • Footnotes for table 4: aAdvanced Nurse Practitioner, bElderly Care Assessment Nurses, cDirect Clinical Care, dAcute Care of the Elderly nurses and eSpecialty Doctors.
  • Footnotes for table 5: aPhysician associates, bDirect Clinical Care and cProactive care of older people undergoing surgery.
  • ISD (Information Services Division) (line 216)

Round 2

Reviewer 1 Report

Thank you for your revisions and for improving the clarity of the manuscript. Now that the appendix are attached and visible to review, I have multiple comments and suggestions that require editing or improving as they both related to the results and to improve clarity for the reader.

3.1 suggest language be improved and that the authors remain consistent throughout the manuscript when reporting... for example line 86: Of these four (57%) offer 24 hour services, 7 days a week. One (14%) operated… and two (29%) have dedicated…

Figure 1: you need to write out 24/7

Line 87 write out 24/7…in the entire manuscript then abbreviate after introducing here, however in tables and  appendix need to always write out for the reader, as these should be able to be read without requiring explanation.  

Line 91 (2: 29% each) confusing, check this and rephrase and rewrite. As you have done in 3.2 line 98 (n=23), then in line 100 (11, 48%). Be consistent. Either write out n=11 or write out 48%, but we do not need both since you add this in the appendix and the reader and look at this for further information and clarification.

Language line 114 “acute take”: is this local language? Do you mean acute admission here?

3.32 Therapy staff: suggest rewriting and keep for example all the physio data together then introduced all the OT data. Too confusing for the reader otherwise.   (lines 126-134)

3.7 One hospital (add %) had a dedicated…

3.8 line 159 consider revising to : Multidisciplinary teams shared notes in 15 hospitals (58%)…

3.92 consider revising to: Of the seven Scottish Health Boards, twelve hospitals (add %)  self reported providing active….

3.93 line 177 Five (add%) Scottish Hospitals…

Line 211 clarify what is decompensated frailty. You have not introduced this before and there is no reference.

Appendix 1 write out all abbreviations either in the text or as a footnote (NHS, DASH, BGS, HIS…) the readers should be able to read the appendix without looking up things

Table 3, 4, 5, Appendix 4, 5, 8: consider changing Hospital Name to Hospital Code, since you throughout the manuscript have given then a number, which is a code, rather than a name.

Appendix 6 write out ED, GP, CGA

Appendix 7 write out WTE, clarify in foot note what are staff grades.

Appendix 8 write out M-F and S&S

Appendix 10 and 11 write out PT and OT and explain for reader what are Band 5&6

Appendix 12 write out abbreviations and or footnote all  

Author Response

06/01/2022

Subject: Re-submission of Manuscript ID: healthcare-1520010

Dear Sir/Madam,

We would like to express our appreciation for the additional feedback that you provided for our paper. It will have significantly improved our paper, “Postcode lottery in healthcare? Findings from the Scottish National Comprehensive Geriatric Assessment in Secondary Care Audit 2019” especially clarifying the appendices.

Please find below our responses to the reviewers’ comments where have made all additions in blue and deletions in red (in the attached document). All changes are also ‘tracked’ on the manuscript according to the guidance of Healthcare journal.

We sincerely hope that the revised manuscript addresses all the issues.

We look forward from hearing from you.

Yours sincerely,

Catriona Young (on behalf of the authors)

Comment to the author: 3.1 suggest language be improved and that the authors remain consistent throughout the manuscript when reporting... for example line 86: Of these four (57%) offer 24 hour services, 7 days a week. One (14%) operated… and two (29%) have dedicated…

Response: Thank you for the suggestion. We feel the proposed alteration makes the results clearer. We further clarified this by stating, “Of these, 4 (57%) run a 24-hour service 7 days a week. One (14%) operated only within normal working hours Monday to Friday, and two hospitals (29%) have dedicated frailty assessment beds within a general ward area (Figure 1).” (Lines 86-89).

Comment to the author: Figure 1: you need to write out 24/7

Response: This has been fixed on Figure 1.

Comment to the author: Line 87 write out 24/7…in the entire manuscript then abbreviate after introducing here, however in tables and  appendix need to always write out for the reader, as these should be able to be read without requiring explanation.  

Response: 24/7 has been expanded as “7 days a week” on line 87 and in Appendix 4.

Comment to the author: Line 91 (2: 29% each) confusing, check this and rephrase and rewrite. As you have done in 3.2 line 98 (n=23), then in line 100 (11, 48%). Be consistent. Either write out n=11 or write out 48%, but we do not need both since you add this in the appendix and the reader and look at this for further information and clarification.

Response: Thank you this was helpful feedback. We have changed it to remain consistent with percentages. Lines 89-93 changed from “The majority (86%) of these units used a form of frailty criteria as a screening tool and also had an age criterion of either greater than or equal to age 65 years or greater than or equal to 75 years (2: 29% each).” to, “The majority (86%) of these units used a form of frailty criteria as a screening tool. Age was another criterion used in the form of either greater than or equal 65 years (29%) or greater than or equal to 75 years (29%). 43% used a combination of frailty and age to screen patients.”

The counts were removed from lines 97-99 so that it is just the percentages that remain “In remaining 19 hospitals, patients admitted with frailty received their initial assessment in either a general medical admissions unit (7, 36%), general medical ward (8, 42%) or a general ward for older adults (4, 21%)”.

The count of 23 hospitals with consultant geriatrician cover was removed so that it reads as “All hospitals with consultant geriatrician cover (n=23) were asked about all possible routes of admission under their care.” (Line 100-101).

In 3.2 Routes of admission, the counts were also removed so that lines 102-105 now reads as, “The most frequent route was via an acute medicine department (11, 48%) or by patients identified by a geriatrician in the acute medicine department (9, 38%); other routes included Emergency Department (8, 35%) and direct GP (General Practitioner) referrals (4, 17%).”

Comment to the author: Language line 114 “acute take”: is this local language? Do you mean acute admission here?

Response: Thank you for this observation of language used. We have changed it from “Geriatricians based in hospitals 4 and 21 had no sessions specifically for the ‘acute take’ of frail older adults at the time of the audit, whereas the highest number of acute takes were reported by hospital number 7 (mean 3.7 acute sessions)” to, “Geriatricians based in hospitals 4 and 21 had no sessions specifically for the ‘acute take’   review of acute admissions of frail older adults at the time of the audit, whereas the highest number of acute takes were reported by hospital number 7 (mean 3.7 acute sessions).” (Lines 116-119).

Comment to the author: 3.32 Therapy staff: suggest rewriting and keep for example all the physio data together then introduced all the OT data. Too confusing for the reader otherwise.   (lines 126-134)

Response: The therapy staffing subheadings have been divided as:

“3.3.2. Physiotherapists (Appendices 9 and 10)

The questionnaire asked about the number of therapists specifically employed to review patients admitted acutely with frailty.  The number of specialist physiotherapists per 10,000 older people aged ≥65 years varies across the Scottish Health Boards with a median of 0.22 [range: 0.0-1.07] physiotherapists dedicated to acute geriatrics medicine per 10,000 population aged ≥65 years. The mean hours spent by physiotherapists reviewing new admissions to geriatric medicine beds were 4.6 hours during weekdays (Monday to Friday) and 1.8 hours at the weekend. 

3.3.3 Occupational therapists (Appendix 11)

The median number of specialist occupational therapists per 10,000 population aged ≥65 years was 0.34 [range: 0.0-2.13]. Similar to that of physiotherapists, the mean hours spent by occupational therapists reviewing new admissions to geriatric medicine beds were 4.7 hours during weekdays and 1.7 hours at the weekend.

We were unable to correlate the number of frailty specific therapists in each unit with the size of the older patient population covered, but there are 8 hospitals without frailty specific physiotherapists and occupational therapists of which only 4 centres are remote and rural.”

Comment to the author: 3.7 One hospital (add %) had a dedicated…

Response: A percentage (4%) has been added (line 161).

Comment to the author: 3.8 line 159 consider revising to : Multidisciplinary teams shared notes in 15 hospitals (58%)…

Response: The sentence reading, “Shared notes were being used by the multidisciplinary team in 15 hospitals (58%)” has been removed. It was replaced with “Multidisciplinary teams shared notes in 15 hospitals (58%).” (Line 166).

Comment to the author: 3.92 consider revising to: Of the seven Scottish Health Boards, twelve hospitals (add %)  self reported providing active….

Response: We have removed “across 7 Scottish Health Boards” so that it reads “There were 12 hospitals who identified themselves as providing active input into orthopaedics (see Table 4).” (Lines 177-178).

Comment to the author: 3.93 line 177 Five (add%) Scottish Hospitals…

Response: The percentage has been added (19%). (Line 184).

Comment to the author:  Line 211 clarify what is decompensated frailty. You have not introduced this before and there is no reference.

Response: We appreciate the feedback and have changed it to, “Geriatric medicine specialists will need to focus their expertise on those with decompensated frailty who are frail, rather than all those in a certain age bracket, with significant implications on demand for geriatric medical care.” (Line 218).

Comment to the author: Appendix 1 write out all abbreviations either in the text or as a footnote (NHS, DASH, BGS, HIS…) the readers should be able to read the appendix without looking up things

Response: Thank for the observation about all the abbreviations used in the appendices that we overlooked. We have expanded on them in Appendix 1 in the footnotes:

aNational Health Service
bAberdeen Centre for Health Data Science
cBritish Geriatric Society
dHealth Improvement Scotland
eSpecial Interest Group

Comment to the author: Table 3, 4, 5, Appendix 4, 5, 8: consider changing Hospital Name to Hospital Code, since you throughout the manuscript have given then a number, which is a code, rather than a name.

Response: Thank you for noticing this. We have replaced Hospital Name with Hospital Code in Table 3, 4, 5, Appendix 4, 5 and 8.

Comment to the author: Appendix 6 write out ED, GP, CGA

Response: We have expanded on the abbreviations in the footnotes:

aEmergency department

bGeneral Practitioner

cComprehensive Geriatric Assessment

Comment to the author: Appendix 7 write out WTE, clarify in foot note what are staff grades.

Response: We have expanded on the abbreviations in the footnotes:

aWhole Time Equivalent

bNational Training Number

cStaff grade are doctors at least two years into specialty training however are not in a consultant training pathway.

Comment to the author: Appendix 8 write out M-F and S&S

Response: We have expanded on the abbreviations in the footnotes:

aMonday to Friday

bSaturday and Sunday

Comment to the author: Appendix 10 and 11 write out PT and OT and explain for reader what are Band 5&6

Response: Physiotherapist has been written out in full and there is a footnote for band 6 and 5:

aSpecialist physiotherapist
bPhysiotherapist

aSpecialist occupational therapist
bOccupational therapist

Comment to the author: Appendix 12 write out abbreviations and or footnote all  

Response: Thank you for this feedback. We have expanded the abbreviations:

aMultidisciplinary Team

bOccupational therapist

cPhysiotherapist
